# Review on Main Arboviruses Circulating on French Guiana, An Ultra-Peripheric European Region in South America

**DOI:** 10.3390/v15061268

**Published:** 2023-05-29

**Authors:** Timothee Bonifay, Paul Le Turnier, Yanouk Epelboin, Luisiane Carvalho, Benoit De Thoisy, Félix Djossou, Jean-Bernard Duchemin, Philippe Dussart, Antoine Enfissi, Anne Lavergne, Rémi Mutricy, Mathieu Nacher, Sébastien Rabier, Stanislas Talaga, Antoine Talarmin, Dominique Rousset, Loïc Epelboin

**Affiliations:** 1Centre d’Investigation Clinique Antilles-Guyane, Inserm 1424, Centre Hospitalier de Cayenne, 97306 Cayenne, French Guiana, France; timothee.bonifay@ch-cayenne.fr (T.B.); paul.leturnier@ch-cayenne.fr (P.L.T.);; 2Infectious Diseases Department, Centre Hospitalier de Cayenne, 97306 Cayenne, French Guiana, France; 3Microbiota of Insect Vectors Group, Institut Pasteur de la Guyane, 97300 Cayenne, French Guiana, France; 4Santé Publique France, Cellule Guyane, 97300 Cayenne, French Guiana, France; 5Laboratoire des Interactions Virus-Hôtes, Institut Pasteur de la Guyane, 97300 Cayenne, French Guiana, France; 6Unité d’Entomologie Médicale, Institut Pasteur de la Guyane, 97300 Cayenne, French Guiana, France; 7Institut Pasteur de Madagascar, Antanarivo 101, Madagascar; 8Laboratoire de Virologie, Institut Pasteur de la Guyane, 97300 Cayenne, French Guiana, France; 9Emergency Department, Centre Hospitalier de Cayenne, 97306 Cayenne, French Guiana, France; 10Unité Transmission, Réservoir et Diversité des Pathogènes, Institut Pasteur de Guadeloupe, 97139 Les Abymes, Guadeloupe, France

**Keywords:** arbovirus, French Guiana, *Dengue* virus, *Chikungunya* virus, *Zika* virus, *Yellow fever* virus, *Mayaro* virus, *Tonate* virus, *Oropouche* virus

## Abstract

French Guiana (FG), a French overseas territory in South America, is susceptible to tropical diseases, including arboviruses. The tropical climate supports the proliferation and establishment of vectors, making it difficult to control transmission. In the last ten years, FG has experienced large outbreaks of imported arboviruses such as *Chikungunya* and *Zika*, as well as endemic arboviruses such as dengue, *Yellow fever*, and *Oropouche* virus. Epidemiological surveillance is challenging due to the differing distributions and behaviors of vectors. This article aims to summarize the current knowledge of these arboviruses in FG and discuss the challenges of arbovirus emergence and reemergence. Effective control measures are hampered by the nonspecific clinical presentation of these diseases, as well as the *Aedes aegypti* mosquito’s resistance to insecticides. Despite the high seroprevalence of certain viruses, the possibility of new epidemics cannot be ruled out. Therefore, active epidemiological surveillance is needed to identify potential outbreaks, and an adequate sentinel surveillance system and broad virological diagnostic panel are being developed in FG to improve disease management.

## 1. Introduction

French Guiana (FG) is a French overseas territory located on the Northeastern coast of South America and is part of the outermost regions of Europe. Due to its geographical location, it is prone to tropical diseases, including both endemic and epidemic arboviruses, but it also benefits from health facilities with European standards. Arthropod-borne viruses (arbovirus) refer to viruses that are maintained in nature through biological transmission between a susceptible vertebrate host and a hematophagous arthropod. Some of these viruses cause zoonoses that rely on a reservoir of non-human animal species for sustenance [1]. Tropical areas have a history of being heavily impacted by arboviruses, as the environment is favorable for the proliferation and establishment of the vectors involved in transmission. In some tropical areas, poor housing conditions or sanitation may increase the level of contact between human recipients and vectors, particularly in urban areas [2]. While the tropical climate may support the growth and diversity of arthropods, the epidemic transmission of arbovirus still depends on the combination of the arbovirus and its competent vector (anthropophilic and urban) [3].

Arbovirus infections occur in two forms: endemic (with sporadic cases) and epidemic forms (often imported). In 2014 and 2016, *Chikungunya* (CHIKV) and *Zika* (ZIKV) viruses were imported to the Americas, including FG, and caused large outbreaks. Millions of people were infected. On the other hand, some autochthonous arboviruses, known for many years, have managed to maintain themselves without ever giving rise to large-scale outbreaks on the French territory, such as the *Oropouche* virus (OROV) or *Mayaro* virus (MAYV). Of course, these manifestations, determined by the distribution and behavior of their vectors, may differ, which makes epidemiological surveillance more complex [4].

During the last decade, seven arboviruses involving human cases have particularly shed light on them in a more or less striking fashion: *Dengue* virus (DENV), CHIKV, ZIKV, *Yellow fever* virus (YFV), MAYV, *Tonate* virus (TONV), and OROV. The objective of this article was to summarize the current and most recent knowledge on these arboviruses in FG focusing on local specificities, and to discuss the future challenges of arbovirus infection epidemiology.

## 2. State of the Art on the Seven Main Arboviruses in French Guiana

### 2.1. Dengue Virus

*DENV* belongs to the *Flaviviridae* family and the genus *Flavivirus*. Currently, *Aedes aegypti* is the primary vector identified for DENV transmission in FG [5]. When looking at the past dengue fever epidemics recorded in FG, the etiology should be considered with a certain distrust; indeed, until the 1980s, dengue diagnosis was based only on clinical manifestations. In fact, it was recently shown that several epidemics described in the 19th century attributed to DENV were possibly due to another arthritis-inducing *Alphavirus* such as CHIKV, the only known pandemic *Alphavirus*, or MAYV [6]. Historically, the first confirmed DENV epidemic occurred in 1969–1970: *Dengue* 2 virus was isolated and serological conversions were noted in three patients [7]. From 1970 to 1977, *Dengue* virus was not isolated. Since then, dengue epidemics have followed with an ever increasing number of cases [8,9]. Since the early 2000s, dengue has transitioned from an endemic to a hyperendemic mode in FG, with the co-circulation of different serotypes. Regular outbreaks occur every 3–5 years, although this periodicity was slightly extended during the successive emergence of CHIKV and ZIKV (Table 1). The dominant serotypes change from one epidemic to another: all the DENV serotypes have been circulating in French Guiana, with a lower representation of DENV4. DENV1 has been predominant in 2009 and 2020–2021, DENV2 in 2013, and DENV3 during the 2005–2006 epidemics. In April 2023, some cases were reported, mainly DENV3 [10]. Notably, the four serotypes of DENV were detected through reverse transcriptase polymerase chain reaction (rtPCR) in wild mammals order Rodentia, Marsupialia, and Chiroptera in sites with and without human dengue cases investigated in FG [11].

During the non-epidemic phases, particularly between 2013 and 2019, reported cases were based on positive anti-DENV IgM only, and no autochthonous case was diagnosed through PCR. This highlights that endemic circulation appears to be very limited because cases diagnosed through IgM serodiagnosis are not confirmed and may reflect past infections with persistent detectable IgM, immunological reactivations caused by a nonspecific acute infectious syndrome, or cross-reactivity with other flaviviruses such as YFV or ZIKV. These cross-reactivity phenomena may be amplified in populations with a history of exposure to flavivirus through vaccinations or infections [12]. In recent years, outbreak frequency has been rising as shown by the occurrence of outbreaks in 2006, 2010, 2013, and 2020–2021. The number of clinically suggestive cases ranged from 10,000 to 18,000 depending on the outbreak. The latest outbreak started in January 2020 and ended in June 2021 and was primarily caused by DENV-1. This outbreak was concurrent with the SARS-CoV-2 pandemic which settled in FG from May 2020. The number of suggestive cases was around 10,891 (against 13,240 in 2013), for an incidence rate of 38/1000 inhabitants. During this outbreak, 282 patients were hospitalized (against 701 in 2013) and 3 deaths were recorded, 2 of which were indirectly linked to dengue (Figure 1). There has probably been a significant underestimation of the number of dengue cases in French Guiana during the 2020–2021 epidemic. Indeed, at that time, the slightest fever was considered to be a SARS-CoV-2 infection until proven otherwise, with extensive protective measures for health care staff which meant that blood samples were only taken on a drip basis. Access to care was limited due to the shutdown of many hospital activities, and the fever pathway to test patients for both dengue and coronavirus disease 2019 struggled to be established. It is difficult to know whether the lockdown has contributed to the dengue epidemic. It can be assumed that patients, forced to stay at home, were more susceptible to attacks by *Aedes* growing in the peri-domestic breeding grounds. In addition, the mosquito control service of the Collectivité Territoriale de French Guiana reports the continuity of activities during the coronavirus disease 2019 period. It is therefore difficult to support a decrease in mosquito control activity, in view of this report, as the consumption of deltamethrin in 2020 was noted higher than in previous years, greater than in 2014 and equivalent to 2016.

The seroprevalence in 2017 was estimated at 73.1% in the general French Guianese population, all serotypes combined [13]. FG, as with many tropical regions, is affected by infectious diseases that share many common clinical and biological features at some point in their natural history, such as *Dengue*, *Chikungunya*, and *Zika* virus infections, malaria, primary HIV infection, Q fever, leptospirosis, and hantavirus. Therefore, a specific biological assessment is mandatory to confirm diagnoses and discard differential diagnoses that would justify specific treatment. Studies have recently shown that a C-reactive protein (CRP) > 50 mg/l is highly suggestive of a diagnosis other than dengue, such as leptospirosis or malaria [14,15]. Rapid diagnostic tests are also useful tools which are already used in remote areas of FG where physicians face unspecified febrile illness and do not have easy access to laboratory facilities. To date, no dengue vaccination program has been implemented in FG yet. A recent statement by the French High Authority for Health has specified that FG is a high endemic area and that DENV vaccination with the live attenuated vaccine DENGVAXIA (SANOFI) can be proposed to immunocompetent people aged between 6 and 45 years old, following the extension in December 2021 of the marketing authorization granted by the European Commission, if a previous DENV infection can be demonstrated to avoid a risk of primoimmunization. However, because no robust and validated test for the diagnosis of past DENV exists yet, there is no recommendation for a large vaccination program in FG, as well as in other overseas French regions. The European Medicines Agency gave a positive opinion on 14 October 2022. A marketing authorization was granted by the European Commission on 8 December 2022 for use in people aged 4 years and older, regardless of previous exposure to the infection. QDENGA is a live attenuated chimeric recombinant vaccine, consisting of four recombinant viruses constructed on the basis of *Dengue* virus 2 and expressing the surface proteins of the four *Dengue* viruses (DEN 1 to DEN 4). During its development, it was shown to be effective in preventing infection and severe forms of the disease requiring hospitalization. The protective efficacy against these severe forms has been estimated at 90.4% and is maintained at over 80% 4.5 years after vaccination. The vaccine was generally well tolerated and no significant risks were identified throughout the trials. Remarkably, no worsening of dengue cases was observed in vaccinated individuals, a phenomenon that led to limitations in the use of the DENGVAXIA vaccine produced by Sanofi Pasteur, licensed since 2018. The vaccine regimen consists of two subcutaneous injections of the QDENGA vaccine 3 months apart. For France, it is now up to the High Authority for Health to specify the place and recommendations for the use of this new vaccine.

### 2.2. Chikungunya Virus

*CHIKV* belongs to the *Togaviridae* family and the *Alphavirus* genus. As DENV, *Ae. aegypti* is the main vector identified in FG. CHIKV infection is characterized by a dengue-like syndrome associated with joint pain and swelling. In French Guiana, the biological diagnosis relies on PCR at the early stage (D0 to D7) and on IgM detection (from D5). Between 2014 and 2015, the American continent faced an unprecedented epidemic of infection by CHIKV. Two lineages were identified during the epidemic: Asian and ECSA lineage; the latter is currently only established in Brazil and Paraguay and has not been identified in FG [16]. After the first autochthonous cases of CHIKV infection reported in Saint Martin (French West Indies) at the end of 2013, the first autochthonous cases from the South American continent were reported in FG in February 2014. The estimated total number of cases recorded by Santé Publique France was about 16,000 cases (Figure 2). No autochthonous case has been reported in French Guiana since April 2015.

In 2021, the seroprevalence in the general French Guianese population was estimated at 20.3% [13]. This gap between recorded cases and the seroprevalence (that estimates around 55,000 people had CHIKV infections) highlights that only 25% of CHIKV infection cases were reported by the surveillance system. This was probably due to asymptomatic or paucisymptomatic cases, cases considered as dengue by clinicians, as well as a low level of healthcare use in the epidemic’s context, or renouncing care, a significant problem in FG [17]. The reported symptoms were relatively similar to those described in La Reunion Island, Indian Ocean, with a picture of high-grade fever and distal joint pain at the onset of the disease. However, atypical and/or severe cases were notified with neurological forms, encephalitis or Guillain-Barré syndrome, septic shock due to CHIKV or thrombotic thrombocytopenic purpura [5,18]. Although CHIKV has not been reported in FG and the West Indies since 2016, apart from rare, imported cases, the risk of a new epidemic in the mid-term remains real. Indeed, Brazil has been facing new epidemics since 2020 with several tens of thousands of cases recorded (mainly due to the ECSA lineage) [19]. Thus, any fever with arthralgia on return from FG should raise suspicion of arbovirus infection and CHIKV should be investigated. In addition, a virus similar to *Chikungunya*, the *Mayaro* virus (see below), which belongs to the same viral genus, has a very similar clinical presentation and is endemic in FG, and must be evoked in the event of a “CHIKV-like” attack.

### 2.3. Zika Virus

The ZIKV epidemic, which began in the Pacific in 2013, spread to Brazil and then to the entire American continent by 2015. After DENV-related outbreaks in 2013 and CHIKV-related outbreaks in 2014–2015, an outbreak due to the emerging ZIKV (*Flavivirus* genus) and transmitted by *Ae. aegypti* occurred in FG, in 2015–2017, where 23% of the population was infected, with only 26% reporting symptoms [20] (Figure 3). ZIKV infection seemed completely benign until 2014, when more worrying forms were described, first in French Polynesia and then in Brazil [21]. These severe forms included neurological disorders in adults, with an over-incidence of Guillain-Barré syndrome [22], and fetal and congenital damage, illustrated by severe microcephaly in fetuses or newborns whose mothers were infected during pregnancy [23]. While the initial data were extremely alarming, it would appear that the burden of ZIKV was lower than initially described. In fact, while significant, with around 15% of children exposed in utero to *ZIKA*, the complication rates appear similar to those of other congenital infections such as cytomegalovirus (CMV) infection [24]. The Saint Laurent du Maroni hospital, in western FG, has been actively sharing their experience on the maternal–infant impact of ZIKV infection [25]. A recently published study, in which 24 of the 546 investigated pregnant women lived in FG, assessed the risk of adverse pregnancy and early childhood likely related to in utero ZIKV exposure and found that among the 555 fetuses and infants, the overall risk was 15.7% (9.4% for mild abnormalities and 3.6% for severe sequelae or fatal outcomes) [26]. In another study assessing the long-term neurological development of children diagnosed with congenital ZIKV infection at birth in FG, 129 children exposed to ZIKV in utero, born and managed in western FG were followed for up to 3 years of life. Compared to those who tested negative at birth, infected neonates had a higher risk of adverse neonatal and early infantile outcomes (aRR 10.1 [3.5–29.0]), neurological impairment, neurosensory alterations or delays in motor acquisition (aRR 6.7 [2.2–20.0]), and suspected neurodevelopmental delay by three years of life (aRR 4.4 [1.9–10.1]) [25]. In 2015–2017, routine serologic screening and enhanced ultrasound surveillance were recommended during pregnancy. In case of confirmed infection, a close postnatal follow-up was implemented. Of note, no case of ZIKV infection has been detected in FG since March 2017 despite regular molecular screening in patients with dengue-like cases. Indeed, ZIKV PCR and/or IgM are systematically screened when a request for an arbovirus diagnosis is sent to the associated National Reference Centre for arboviruses located at the Institut Pasteur in French Guiana in Cayenne. However, it is no longer systematically sought in the three main hospitals of Cayenne, Kourou and Saint Laurent du Maroni. No imported case has been reported, and, according to our knowledge, currently the ZIKV does not appear to be circulating in neighboring regions.

### 2.4. Yellow Fever Virus

YFV can lead to a serious infection manifested by a hepatorenal hemorrhagic syndrome. It is a *Flavivirus* native to Africa and imported through triangular trade and circulating in both Africa (95%) and South America where it has become endemic [27]. It can be responsible for large epidemics, for example, those in Brazilian Southeast states, such as Minas Gerais, Espírito Santo, Rio de Janeiro, and São Paulo in 2016–2018, or more recently in 2021 in Venezuela [28,29]. An international certificate of immunization against *Yellow fever* is required for residents of FG and travelers wishing to go there. The certificate is valid for life after a single injection (WHO 11 July 2016), with the exception of immunocompromised persons, persons traveling to a country where active circulation of the virus is reported, women who have been vaccinated during pregnancy, and children over the age of 6 who received their first injection before the age of 2 [30]. The vaccine is recommended from 9 months of age for children traveling to or living in countries at risk. In French Guiana, the yellow fever vaccine Stamaril^®^ (Sanofi Pasteur) is used. In 2017, a vaccine coverage survey was conducted among 2697 individuals from the 22 municipalities of FG. YFV vaccination coverage was estimated at 95.0% (95% CI: 93.4–96.2) but with spatial heterogeneity. The lowest coverage levels were in the western part of the territory along the Surinamese cross-border region, particularly in children not enrolled in school, immigrant adults, and people with low resources [31].

There is no proof that yellow fever occurred in FG before 1763, and thus the introduction of the virus was through the slave trade with the forced arrival in the Americas of people coming from Africa [27]. The first report of “vomito negro” appears during the Kourou expedition in 1763. From 1763 to 1888, yellow fever was reported repeatedly [32]. Epidemics were observed in 1765, 1793–1796, 1802, 1850, 1855 (with 27.2% mortality), 1873–1874, 1876, 1877, and 1885. In 1902, an epidemic started in Saint-Jean-du-Maroni and reached Cayenne in a few months: 471 cases were recorded with 139 deaths. *Ae. aegypti* was likely the main vector during these epidemics but the roles played by *Haemagogus* and *Sabethes* during a sylvatic cycle should not be discarded. No other sylvatic *Aedes* species have been implicated in the sylvatic transmission of YFV in our territory. After that, no human cases of yellow fever were reported from 1902 to 1998, when the first fatal case was reported in a Wayana Amerindian woman in the upper Maroni River [33]. No cases were reported for 20 years until 2017. Four cases have been reported since then: two cases in 2017 and 2020 in Brazilian illegal gold miners working in forest camps, one case in 2020 in a Wayana Amerindian teenager from the upper Maroni River who was vaccinated against yellow fever in childhood but did not receive any booster thereafter and was co-infected with coronavirus disease 2019, and, finally, one case in 2018 in a Swiss citizen who was not vaccinated [34,35]. All of these cases were fatal, which suggests that the circulation of this virus is under-detected, due to its sylvatic cycle, and that milder cases are not detected and reported.

The zoonotic circulation of YFV has been documented in FG, although such studies are rare and date back to the 1990s. A wildlife rescue operation at the Petit Saut hydroelectric dam in 1994–1995 involved collecting blood samples from 574 individuals of 27 species, under veterinary control. The results showed that 10 species of various animal orders had yellow fever sero-neutralizing antibodies, including golden-handed tamarin, white-faced saki, and red howler monkeys (primates), agouti and porcupine (rodents), peccary (artiodactyl), tayra (carnivore), two-toed and three-toed sloths, and anteater (xenarthrans). Among the species tested, howler monkeys had the highest exposure to YFV, with a sero-neutralizing antibody prevalence of 18%. All of the infected animals were adults, and no obvious clinical effects were observed [36].

### 2.5. Mayaro Virus

MAYV is an *Alphavirus* of the *Togaviridae* family related to CHIKV and was first described in Trinidad in 1954. Its main vector is a sylvatic *Haemagogus* mosquito but *Ae. aegypti* has also been implicated in transmission to the human host. *Ae. aegypti* is a competent vector of MAYV under laboratory conditions [37], and has been found naturally infected in parks and gardens within urban areas [38]. Research has also shown that *Aedes albopictus*, another important human disease vector, is able to transmit MAYV to mice in the laboratory [39]. Other mosquito species from the genus *Mansonia, Culex, Sabethes, Aedes*, and *Psorophora* have also been implicated as potential vectors of MAYV [40]. MAYV has circulated in Latin America, causing several outbreaks in the Amazon region of Venezuela, Peru, Bolivia, and Brazil. It was first isolated in FG in 1996 [41]. A country-scale study recently showed a seroprevalence ranging from 1% in Cayenne to 23.5% in some isolated communes of the upper Oyapock River and upper Maroni River [13]. A retrospective study identified 17 human cases between 2003 and 2019, mostly acquired in the deep forest [42]. The clinical and biological picture was similar to CHIKV infection with fever and arthralgia. One patient had acute meningoencephalitis, and four had persistent arthralgias. On 13 October 2020, the French health authorities officially reported 13 laboratory-confirmed cases of Mayaro fever in FG. In September 2020, the Institut Pasteur de la Guyane (IPG) (associated laboratory of the French National Reference Center for arboviruses) identified two cases of MAYV infection in patients with joint paints confirmed by rtPCR and one probable case found positive for MAYV antibodies. rtPCR was positive in 13 out of 97 tested samples collected from patients with dengue-like symptoms between 15 July and early October. The onset of symptoms for the 13 confirmed cases ranged from 18 July to 29 September 2020. Of the 13 confirmed cases, 11 lived in the urban coastal area including 9 from the Cayenne area (Cayenne: 1, Rémire: 4, Matoury: 4), 1 from Kourou, and 1 from Montsinery-Tonnegrande. Only two cases lived in a rural/sylvatic area, both in Roura (including one in the village of Cacao, located further in the interior). The age of these cases ranged from 11 to 68 years old (median = 40 years old) and the male to female sex ratio was 1.2:1. The detection of 13 confirmed cases within less than 3 months was thus unusual, as well as the transmission in urban settings in 11 out of 13 (85%) of the identified cases. MAYV must therefore be evoked in the presence of febrile arthralgia in patients living in or returning from Latin America [43]. This arbovirus had also been mentioned as being a virus with potential epidemic bursts [44].

### 2.6. Tonate Virus

TONV is also an *Alphavirus* of the *Togaviridae* family. This virus is regularly detected in FG and belongs to sub-type IIIb of the Venezuelan equine encephalitis virus complex. It was first described in 1973 in FG, in a bird, the crested oropendola (*Psarocolius decumanus),* in the village of Tonate, near Cayenne, and then found in several species in FG and Suriname, including mosquitoes (*Anopheles*, *Culex)* and phlebotomine sand flies (*Lutzomyia)* [45]. It has been isolated in specimens of Cliff Swallow nest bugs (*Oeciacus vicarius)* and in nestling House Sparrows (*Passer domesticus*) and Cliff Swallows (*Petrochelidon pyrrhonota*), which are nestling birds in Colorado and South Dakota [46]. More recently, a study allowed the detection of TONV neutralizing antibodies in four species of bats and the isolation of a TONV strain from a fringe-lipped bat (*Trachops cirrhosis*) serum sample [47]. Although there are a few human publications in Guianese patients, the first of which were two cases reported in 1973 and 1975, this human infection has never been reported outside FG. Two serological studies in the population, one in the 1970s and the other in the 1990s, showed average seroprevalence rates of around 11–14%, with very wide geographical variations from 0 to 35%, the highest rates being found on the coastal plains [48,49]. A retrospective study was carried out on 45 cases identified at the National Reference Center for Arbovirus that were managed in Cayenne Hospital and health centers in remote areas between 2003 and 2016 [50]. The infection mainly affected young men and the symptoms most frequently observed were fever, chills, headache, and diffuse pain. As with MAYV, the biological work-up was not very specific, with lymphopenia found in about 20% of cases, and a CRP above 50 mg/L in 20% of cases. An acute meningoencephalitis with pleomorphic fluid associated with hyperproteinorachia at 1.52 g/L, and a spontaneously favorable evolution was reported. No deaths were reported. The only serious case in the literature is a fatal encephalitis in a 2-month-old child published in 1998 [50]. Recently, the Saint-Laurent-du-Maroni team reported for the first time a case of vertical transmission of the *Tonate* virus in a pregnant woman in FG. The fetus presented severe necrotic and hemorrhagic lesions in the brain and spinal cord [51]. TONV should therefore be considered in the presence of a dengue-like picture, as well as in the presence of an undocumented central nervous system infection.

### 2.7. Oropouche Virus

OROV is an arbovirus of the *Peribunyaviridae* family first identified in humans in 1955 in Trinidad and Tobago and usually transmitted by biting midges of the genus *Culicoides*. Several epidemics have been reported in Latin America, particularly in Brazil, Peru, and Ecuador [52]. In August and September 2020, in the midst of the coronavirus disease 2019 epidemic, around 50 cases were reported for the first time in FG, among the inhabitants of the small village of Saül, located in the heart of the Amazonian forest, with an estimated attack rate of between 43 and 61% [53]. The symptomatology was aspecific, accompanied by fever, headache, and diffuse pain, rarely leading to serious cases. This OROV epidemic resulted in hospital admission for three people, including one for acute lymphocytic meningitis; all of them had a favorable outcome. The reason for the emergence of this arbovirus is not known. One of the advanced hypotheses is the possible presence of Brazilian gold panners, originating from Brazilian states where OROV circulates more regularly, and is present in greater numbers around the village, due to the interruption of anti-panning patrols because of the confinement linked to coronavirus disease 2019. Nevertheless, this virus circulates regularly in Brazil, which indicates a well-established sylvatic cycle; this occurrence cannot be excluded in FG in areas far from inhabited zones, with possible spillover being less frequent than in Brazil. The entomologist’s initial investigation following the outbreak did not allow the identification of the potential vector of this cluster. Only one individual of the main known vector, *Culicoides paraensis* (Goeldi, 1905), was found during this entomological investigation, which was not performed immediately after the epidemic. At that time, numerous *Culex quinquefasciatus* (Say, 1823) were found, and suspected as vectors of this arbovirus. Indeed, although the main vector of OROV is *Cx. paraensis*, mosquitoes of the genus *Culex* and notably *Cx. quinquefasciatus*, which is very present on the littoral, have shown a slight degree of vectorial competence in the laboratory. Finally, two years later, at the same season as the outbreak, the presence of abundant *Culicoides paraensis* was proven within the village (Institut Pasteur de la Guyane’ works), confirming its probable past vector role [54].

## 3. Challenges for the Future

### 3.1. Increase in Factors Favoring Arbovirus Epidemics

The population of FG is increasing rapidly with a birth rate of 3.53 children per woman (vs. 1.8 children per woman in France). Thus, combined with growing poverty (53% of people living below the poverty line in FG in 2020) and a massive urbanization of FG, this demographic expansion will be challenging for the public authorities to avoid, or at least limit the expansion of anthropophilic vectors such as *Ae. aegypti.* Therefore, specific actions to reduce the number of larvae breeding sites, promote access to drinking water in order to limit water storage, offer efficient household waste collection services, and improve the awareness through efficient educational programs within communities or at school are important [55]. We can fear regular epidemics due to the decrease in the collective immunity of the population below the thresholds, allowing the occurrence of epidemics. Moreover, FG has a highly mobile population making the risk of introducing new arboviruses into the country real, as we have experienced in recent years [56]. It is important to identify them in order to pursue an effective and adapted health watch.

### 3.2. Despite Common Characteristics, a Great Heterogeneity

“Cosmopolitan” arbovirus epidemics reported in FG in the last decade have all shared common characteristics. Thus, DENV, CHIKV, and ZIKV were transmitted by a common vector, *Ae. aegypti*, which is an urban, anthropophilic vector. The distribution of the latter in FG is wide and expanding and present in almost all FG locations where people live (except for Camopi and Trois-Sauts) (Figure 4). Rural communities within the FG territory were spared the presence of *Ae. Aegypti* until the mid-2000s, when the first DENV cases were described [57]. A recent seroprevalence study has shown that the populations living along the northern part of the Maroni banks, which were safe from these arboviruses for a long time, have been strongly affected during the past years by DENV, CHIKV, and ZIKV [13,57]. On the other hand, the older arboviruses likely established for centuries on the territory (MAYV, TONV, and YFV) seem to have more varied transmission cycles (sylvatic and urban) and animal reservoirs that are still not well known. The vectors are thus affected and seem to be more diverse (*Aedes, Haemagogus* for MAYV and YFV, *Anopheles, Culex* for TONV). This heterogeneity may impact the dynamics of new epidemics caused by previously reported arboviruses, such as DENV, but also by new introductions and emerging arboviruses. Finally, this heterogeneity of the vector distribution is probably one of the main drivers of the arbovirus seroprevalence heterogeneity observed recently in FG [13].

After the CHIKV and ZIKV major epidemics that followed their introduction to the Americas in 2014 and 2015, respectively, the active surveillance of these arboviruses was maintained which confirmed that there was no detectable current endemic circulation of these viruses. It is worth remembering that CHIKV has been established in Brazil since 2013 and always manifests itself in epidemic form [58]. The risk of endemicity should be considered.

Like many infectious diseases, vector-borne diseases have a strong impact on the most vulnerable populations. Indeed, poverty has been described as a risk factor for ZIKV and CHIKV infection, particularly in the early stages of an outbreak [59,60]. This population is often more exposed to vectors and remote from prevention campaigns or actions. Specific actions targeting this at-risk population have been settled, such as the WASH project developed by the French Red Cross to fight against waterborne and vector-borne diseases in disadvantaged areas [61].

### 3.3. Vector Control

The breeding site removal to prevent larvae and the spatial and domiciliary insecticide spraying to prevent adult mosquitoes are historically the foundations of vector control [62]. Concerning the dengue vector, *Ae. aegypti*, several other strategies are used such as governmental vector control programs, enhanced community engagement, or individual protection campaigns. While several insecticidal molecules and application methods have been used over the last 60 years to maintain the efficiency of vector control, FG, as the French department in the Americas, follows European regulations, which is very unique in South America [63]. As such, only pyrethroids compounds are available against adult mosquitoes, contrary to other South American countries where organophosphates and carbamates can be used. Deltamethrin is the sole insecticide available to spray against adult *Ae. aegypti* in FG, while *Bacillus thuringiensis* var *israelensis* H14 (*Bti)* treatments along with the mechanical removal of *Ae. aegypti* breeding sites are used to reduce larval densities.

The intensive and successive use of several families of insecticides these last decades (*i.e., organophosphates* and pyrethroids) has led to the resistance of mosquito populations, especially *Ae. aegypti*, reducing the efficiency of the treatments. The high resistance to insecticides in mosquitoes [64] as well as the description of the adverse effects of most of insecticides, such as impact on biodiversity and cancerogenic effects [65], have led to the need for alternative methods to be tested in FG, such as *wolbachia*-infected mosquitoes and sterile male methods [66].

While *Aedes* is the major vector of arboviruses, and the only vector involved in large-scale epidemics, particular attention should be paid to *Culex* and *Haemagogus*, the vectors involved in the transmission of autochthonous arboviruses.

It is important to keep in mind that most of the vector control measurements mentioned are only effective in urban settings where mosquito populations exploit artificial habitats for breeding. Those measurements don’t extrapolate to sylvatic settings where mosquito habitats and behavior differ. Therefore, antibody surveillance is essential in human populations and sentinel surveillance may help to detect early epidemics of new or incoming arboviruses.

### 3.4. Risks of Introduction of Aedes Albopictus in French Guiana

*Aedes albopictus* (Skuse, 1894) is one of the most invasive mosquito species in the world. In the last decades, this species has spread from its native range in Asia to the most temperate and tropical inhabited areas of the world, including the Americas [67]. So far, the geographical area known as the Guiana Shield has been largely saved from the spread of the Asian Tiger mosquito. However, in recent years, this species has been documented in the main island of Trinidad and Tobago [68] and in the Brazilian city of Macapá in the state of Amapá [69,70]. The location is only an 8 h drive (600 km) from the eastern border of FG, where the introduction of *Ae. albopictus* is likely to happen through human and trade interactions. This event is likely to occur and remaining questions are when and where [71]. Currently, entomological surveys dedicated to the early detection of *Ae. albopictus* in FG are conducted by the *Direction de la Démoustication et des Actions de Santé* (DDAS) at the international Cayenne—Félix Eboué Airport, the major maritime ports, and the main human settlements situated at the borders with Suriname and Brazil. The establishment of *Ae. albopictus* in FG would probably lead to interspecific larval competition with *Ae. aegypti* which share the same environmental niches [72]. As already suggested in North and South America, the introduction of *Ae. albopictus* could result in geographical displacement and the population reduction in *Ae. aegypti* [73]. Given the differences in vectorial competence between the species, a modification in the overall transmission dynamics and patterns of DENV, CHIKV, and ZIKV in urban areas is possible but would need to be assessed locally. The higher adaptability of *Ae. albopictus*, especially to forested areas near human dwellings, could facilitate the movement of rare zoonotic arboviruses from rural to urban areas, thereby increasing the risk of new infectious diseases emerging in FG.

### 3.5. Zoonotic Arboviruses

Concerning the MAYV, the reservoir and the vector seem well identified in FG. So far, their location in primary forest limits the risk of transmission to humans. However, MAYV belongs to the same family as the CHIKV, which went from being a forest virus to an urban virus. Such a possibility should not be ruled out for MAYV, as well as an adaptation to an animal reservoir closer to humans. Seroprevalence surveillance studies in domestic animals should therefore be regularly performed to assess any change. Similarly, the vector competence of *Ae. aegypti* and possibly *Ae. albopictus* against MAYV should be regularly performed on newly isolated strains or strains whose genome differs significantly from older strains. TONV should be tested regularly on the same principles. The vectors of the OROV certainly need to be better explored in FG in order to better assess the risk for the population.

### 3.6. Potential Emergence of New Arbovirus

In 1966, virology became a crucial aspect of the activities at the Institut Pasteur in FG. The U79 INSERM Group was established in 1968 to conduct research on arbovirus diseases [8]. The group carried out investigations (Figure 5) on YFV and identified several viruses, some of which were already known while others were previously unknown in mosquitoes, wildlife, birds, and humans such as *Tonate*, *Mucambo, Cabassou, Una*, and *Aura* viruses in the *Togaviridae* family, or *Saint-Louis Encephalitis* and *Ilheus* viruses in the *Flaviviridae* family [8,45,48,74,75,76]. Nevertheless, the most frequently identified virus family was the *Peribunyaviridae* family with the *Murutucu* virus; *Oriboca* and *Caraparu Orthobunyaviruses* from the serogroup C; *Guama*, *Bimiti*, and *Catu Orthobunyaviruses* from the Guama group; *Inini* virus from the Simbu group; *Guaroa Orthobunyavirus* from the California group; and *Maguari* and *Wyeomyia Orthobunyaviruses* from the Bunyamwera group. Finally, a few ungrouped viruses, *Itaporanga* and *Aruac* viruses, have been detected [77]. At the time of detection, *Culex portesi* appeared to be the mosquito involved in the sylvatic circulation of the largest number of arboviruses in the region. For each virus, details on the host, the number, and the date of isolation are presented in Table 2 below with the reference of the corresponding publication. 

At least some of them have been anecdotally detected in humans [45]: *Mucambo* virus infected four laboratory workers who reported a sudden onset of high fever, myalgia, headache, nausea or vomiting, and very high asthenia lasting 15 days, after handling the reference strain in 1973; *Aura* virus was incriminated in the infection of a woman with high fever and severe jaundice who died quickly after admission in 1974; *Ilheus* virus was detected in a patient suffering from fever, headaches, and chills in the 1970s; and *Murutucu* virus (group C) was described in three patients with fever, headache, and myalgia in 1973. The U7 9 INSERM Group ended its activities in 1979, and, after that, the focus shifted to dengue fever surveillance until recently. This explains the high number of new arboviruses reported or discovered during this period. The risk of emergence of autochthonous arboviruses is real and aggravated by the rapid urbanization of the FG (destruction of ecosystems, vulnerability to new vectors, or emergence of new vector reservoirs).

The risk of introduction of new arboviruses is not negligible in FG, and may come from either Brazil or from the North-East of South America (*Venezuelan equine encephalitis* virus…). The development of multiplex tools to target all arboviruses, which is most likely to be introduced in the territory, is certainly a major development axis for the coming months in order to respond as soon as possible in the case of febrile syndrome that is not diagnosed with conventional tools. The vectorial competence of the vectors most present in FG should also be tested with respect to a certain number of viruses prevailing in neighboring countries.

### 3.7. Potential Introduction of Arboviruses from French Guiana to Europe and the West Indies

Some of the viruses reported here have a real potential for introduction in Europe. France is particularly at risk because of its special links with its French territories in the Americas, the French West Indies, because of the numerous population exchanges, especially for tourism, and FG to a lesser extent. The risk for the introduction of dengue in Europe is no longer needing to be demonstrated due to the ever-increasing presence of *Ae. albopictus* throughout Europe. The French experience of the summer of 2022 is particularly telling, as the number of reported autochthonous cases in metropolitan France was greater than the total number of cases reported over the period 2010–2021 [83]. Nevertheless, the cases identified in 2022 did not originate from the French territories of America but instead mainly Cuba, the Ivory Coast, and Mexico. CHIKV and ZIKV are also known to have the potential for introduction in Europe, but the current absence of epidemics in FG and the French West Indies can limit the risk. Furthermore, while autochthonous cases have been described in Europe for CHIKV, this has not been reported for ZIKV to our knowledge. It should be noted that some European scientific societies still recommend additional restrictions in the case of medically assisted reproductive assessments in people from Latin America, despite the disappearance of the epidemic more than 3 years ago. To our knowledge, no case of TONV has been reported out of FG, and no case of OROV out of Latin America. On the other hand, several cases of MAYV infection have been described in travelers from Latin America, and, in particular, from FG. Although there is a potential for transmission of this virus by *Ae. aegypti*, no autochthonous case has ever been described outside endemic areas. However, it is important to remain vigilant about this virus that some experts consider as having a pandemic potential.

Concerning the West Indies, exchanges occur daily between FG and the French West Indies, and, through them, with all the West Indies including Haiti. Therefore, some viruses may strike the West Indies as well as FG because of the presence of *Ae. aegypti* (DENV, CHIKV, ZIKV); however, the others have not been described, with the exception of MAYV in Haiti (its autochthonous origin remains to be demonstrated). However, it is important to remain vigilant, especially for YFV, since the competence of *Ae. aegypti* strains from the French West Indies has been demonstrated. The risk of an urban epidemic following the introduction of a YFV cannot be ruled out. Vaccination of the inhabitants of the West Indies before a stay in FG should therefore remain mandatory.

## 4. Conclusions

French Guiana is home to a high diversity of infectious and tropical diseases, illustrated here by several arboviruses. The nonspecific clinical picture implies a good knowledge of the local epidemiology but also an easy access to advanced biology to catch the correct bug. Many of the arboviruses described here have a significant epidemic potential within FG but also abroad. The example of FG allows us to distinguish the viruses with endemic circulation (YFV, MAYV and TONV) from viruses with an epidemic form (DENV, CHIKV, ZIKV) and that are transmitted by *Ae. aegypti*. The resistance of *Ae. aegypti* to insecticides makes control very difficult, and the development of alternative methods, and effective vaccines, is urgent.

Concerning the risk of relapse of certain arbovirus epidemics, can we be reassured today that CHIKV and ZIKV will not return, given that herd immunity or any other unclear phenomenon is sufficient for no epidemic to return for a generation? It cannot be said that a seroprevalence of 20%, like that of CHIKV in FG, provides collective protection. A new epidemic circulation cannot be ruled out, even in the relatively near future, hence the need for surveillance.

The origin of the OROV epidemic remains unknown and underscores the need for active epidemiological surveillance, in FG as well as throughout the Amazon basin. Furthermore, older publications indicate that many poorly known arboviruses may be circulating in the area, causing infection in humans, although they have not been identified in recent decades, due to the lack of an adequate sentinel surveillance system and a sufficiently broad virological diagnostic panel. Such a surveillance network is currently being set up in FG under the coordination of health authorities, with sentinel doctors, private and hospital laboratories, and the NRC for arbovirus of the IPG and with a diagnostic algorithm extended to other arboviruses such as several orthobunyaviruses.

## Figures and Tables

**Figure 1 viruses-15-01268-f001:**
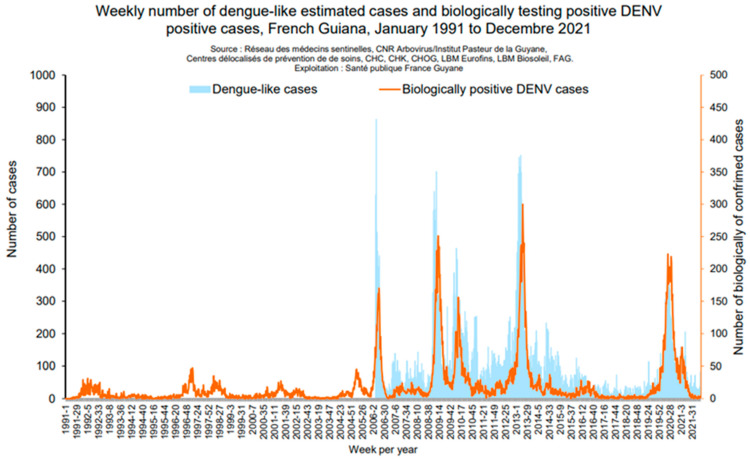
Epidemic curves for DENV epidemics (*Santé Publique France*).

**Figure 2 viruses-15-01268-f002:**
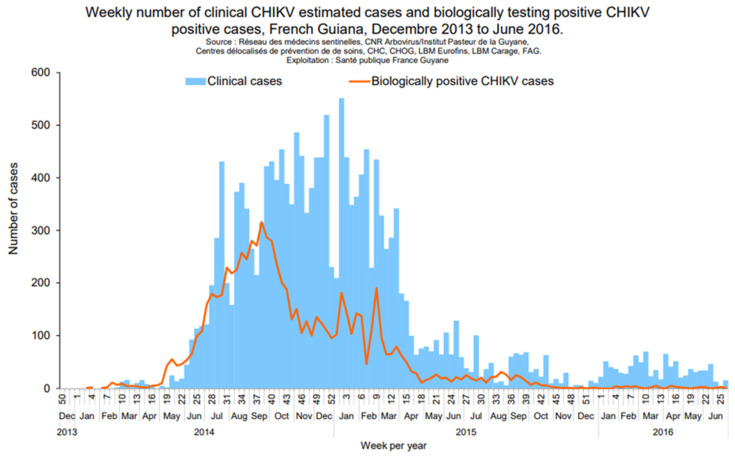
Epidemic curves for CHIK epidemics (*Santé Publique France*).

**Figure 3 viruses-15-01268-f003:**
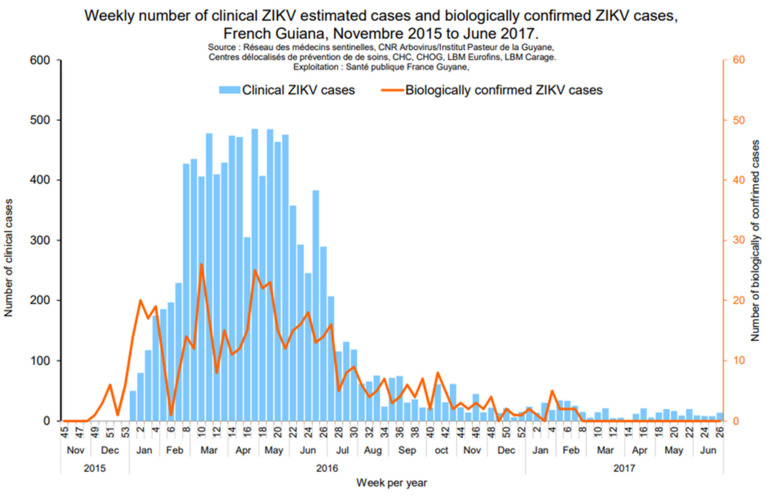
Epidemic curves for ZIKV epidemics (*Santé Publique France*).

**Figure 4 viruses-15-01268-f004:**
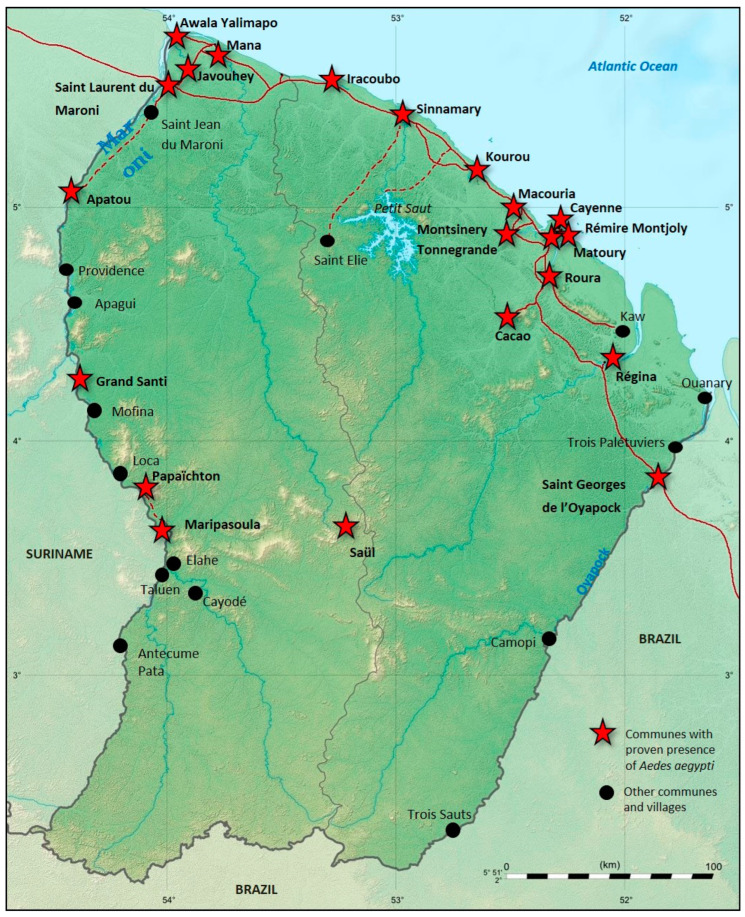
Mapping the distribution of *Ae. aegypti* in French Guiana. Map produced by combining DDAS data with GBIF data (Page et al.). Red cross indicates the presence of *Ae. aegypti*.

**Figure 5 viruses-15-01268-f005:**
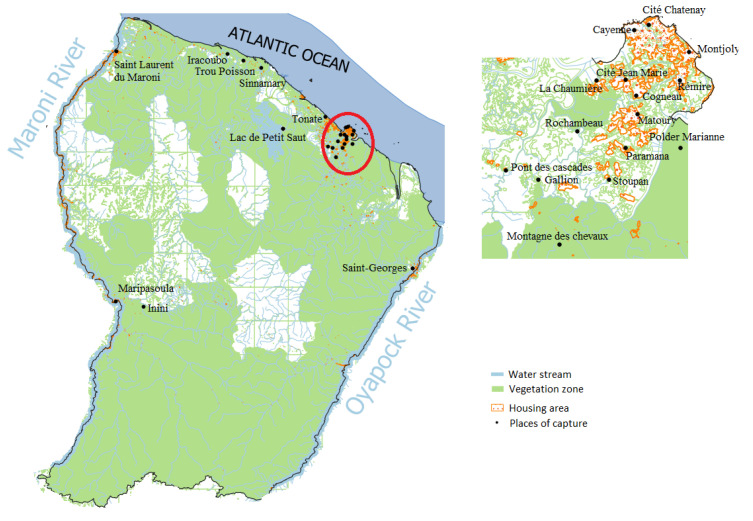
Map illustrating the distribution of mosquito sampling locations during investigations carried out on arboviruses, presented in Table 2.

**Table 1 viruses-15-01268-t001:** Summary of seroprevalence and incidence data for the 7 main arboviruses of interest in French Guiana.

Arbovirus (Acronym)	Family (Genus)	Main Vectors	Sero- Prevalence	Epidemiological Data (Number of Cases)
*Yellow fever* (YFV)	*Flaviviridae* *(Flavivirus)*	*Haemagogus* and *Sabethes * +/− *Aedes aegypti*	95.0%	1 in 1998 2 deaths in 2017–2018 2 deaths in 2020
*Dengue* (DENV)	*Flaviviridae* *(Flavivirus)*	*Ae.aegypti*	73.1%	2005–2006 (DENV3 > DENV2): 13,700–16,200 2009 (DENV1 > DENV 4): ~7800 2013 (DENV2 > >DENV4): ~16,000 2020–2021 (DENV1 > >DENV2): ~10,000
*Zika* (ZIKV)	*Flaviviridae* *(Flavivirus)*	*Ae.aegypti*	23.3%	2015–2017: ~9700
*Chikungunya* (CHIKV)	*Togaviridae * *(Alphavirus)*	*Ae.aegypti*	20.3%	2014–2015: ~16,000
*Mayaro* (MAYV)	*Togaviridae * *(Alphavirus)*	*Haemagogus* spp. *Ae.aegypti*	3.3%	17 from 2003 to 2019 ~15 in 2020
*Tonate* (TONV)	*Togaviridae* *(Alphavirus)*	*Culex portesi*	11.9%	45 from 2003 to 2016
*Oropouche* (OROV)	*Peribunyaviridae* *(Orthobunyavirus)*	*Culicoides* spp. *Culicoides paraensis* *Culex quinquefasciatus?*	ND	41 to 58 in August September 2020 in Saül

**Table 2 viruses-15-01268-t002:** Summary of knowledge on identification of rare arboviruses from publications in French Guiana by virus isolation, molecular biology, as well as common arboviruses in the wild fauna.

Genus	Virus	Reference	Date	Number of Isolations	Place	Vector	Vertebrate Hosts	Humans	Comment
*Alphavirus*	***Mucambo* virus**	[45]	1972	1	Montsinéry	**Phlebotominae** *Lutzomyia* sp.			
		[8,45]	1973	5	Maripasoula		**Birds** *Monasa atra Tachyphonus cristatus* *Trogon violaceus* *Turdus nudigenis*		
		[45]	NA	NA	NA	**Culicidae** *Aedes* sp. *Culex portesi* *Culex* sp. *Haemagogus* sp. *Mansonia* sp. *Sabethini* sp. *Wyeomyia* sp.	**Primates** *Cebus apella* (used as sentinel)		
		[45]	NA	NA	Maripasoula		**Birds ** 4 birds sp.		
		[45]	1972	NA	Cayenne	**Phlebotominae** *Lutzomyia* sp.			
		[45]	1973	3	Cayenne			Four laboratory workers who had handled the reference strain. Symptoms: sudden onset of high fever, myalgias, headaches, nausea or vomiting, very high asthenia lasting 15 days	Since March 1973, it has never been isolated again and seems to have been replaced by another virus of subtype III, *Tonate*
	***Tonate* virus**	[8,45,74]	1973	NA	Tonate	**Culicidae** (n = 71 from 1973 to 1978) *Anopheles* sp. *Coquillettidia* sp. ***Culex portesi*** *Mansonia* sp. *Uranotaenia* sp. *Wyeomyia* sp. **Phlebotominae** *Lutzomyia* sp.	**Birds** *Psarocolius decumanus*	See main paragraph	
		[45]	1973	NA	Gallion Paramana Stoupan Trou Poissons	**Culicidae** *Culex portesi* *Mansonia titillans*			
		[45]	1975	NA	Cité Chatenay Gallion La Chaumière Matoury Paramana Tonate Trou Poissons	**Culicidae** *Coquillettidia venezuelensis* *Culex portesi* *Culex spissipes* *Culex zeteki* *Wyeomyia melanocephala* *Wyeomyia occulta*			
		[45]	1973–1977	16 (birds) 72 (mosquitoes)	Gallion La Chaumière Matoury Paramana Sinnamary Tonate Trou-Poissons	**Culicidae** *Anopheles braziliensis* *Anopheles mediopunctatus* * *Coquillettidia albicosta* (as species of *Culex*) *Coquillettidia venezuelensis* *Culex nigripalpus* *Culex portesi* *Culex spissipes* *Culex zeteki* *Mansonia pseudotitillans* *Mansonia titillans* *Uranotaenia geometrica* *Wyeomyia melanocephala* *Wyeomyia occulta* *Wyeomyia pseudopecten* **Phlebotominae** *Lutzomyia* sp.	**Birds** *Ardeola ibis* *Chiroxiphia pareola* *Elaenia chiriquensis* *Glyphorynchus spirurus* *Leucopternis albicollis* *Myiozetetes cayanensis* *Nycticorax violacea* *Oxyura dominica* *Psarocolius decumanus* *Ramphocelus carbo* *Sakesphorus canadenses* *Sporophila lineola* *Tachyphonus rufus* *Tolmomyias poliocephalus* *Turdus nudigenis* **Rodents** Sentinel mice		
		[47]	2011	1			**Bats** *Trachops cirrhosus*		Isolation of TONV
	***Cabassou* virus**	[45]	1972	1	Paramana	**Culicidae** *Culex portesi*			(n = 15 from 1974 to 1980)
		[8]	NA	NA	Saint Laurent du Maroni		**Bats** *Chiroptera* sp.		
		[8,74,78]	1973–1980	2 (birds) 3 (mammals)	Cité Jean-Marie¤ Gallion La Chaumière Matoury Montagne des chevaux Paramana Tonate Sinnamary Trou-Poissons	**Culicidae** *Anopheles peryassui* *Coquillettidia venezuelensis* *Culex nigripalpus* *Culex portesi* *Limatus pseudomethysticus* *Mansonia titillans* *Wyeomyia occulta*	**Birds** *Tolmomyias poliocephalus* *Turdus nudigenis* **Marsupials** *Didelphis marsupialis* *Philander oppossum* **Rodents** Sentinel mice **Bats** *Chiroptera* sp.		
		[45]	1974–1975	2	Gallion	**Culicidae** *Culex portesi*			
	***Una* virus**	[8,45]	1973	1	Montjoly	**Culicidae** *Psorophora ferox*			**(n = 5 from 1973 to 1977)**
		[45]	1973	1	Sinnamary	**Culicidae** *Coquillettidia venezuelensis*			
		[45]	1973	1	Paramana	**Culicidae** *Coquillettidia albicosta*			
		[8,45]	1975	2	Maripasoula	**Culicidae** *Psorophora lutzii*			
		[78]	1977	NA	Cité Jean-Marie¤	**Culicidae** *Anopheles nimbus*			
			1973	1	Inini		**Birds** *Campephilus rubricollis*		
	***Aura* virus**	[8,78]	1973	2	Inini	**Culicidae** *Aedes serratus*			
		[45]		NA	Polder Marianne	**Culicidae** *Mansonia titillans*			
		[45]	1974	1	Cayenne			One woman with high fever and severe jaundice who died quickly after admission.	The definite proof that the virus was the cause of disease is lacking.
*Flavivirus*	***Ilheus* virus**	[8,45]	1973	3	Iracoubo Maripasoula Montsinéry		**Birds** *Leistes militaris* *Molothrus* *bonariensis * *Momotus momota* *Piaya minuta*		
		[78]	1977	1	Gallion	**Culicidae** *Coquilletidia venezuelensis*			
		[45,79]	1973	2	Macouria				From the blood of a patient suffering a “dengue-like” syndrome, with fever, headaches, chills and co-infection with *Plasmodium falciparum*
	***Saint-Louis Encephalitis* virus**	[8]	1967	1	NA	**Culicidae** *Culex* sp.			
		[8]	1977	1	NA		**Birds** *Anhinga anhinga*		
	***Dengue* virus**	[11,80]	2006	5	Saint Georges de l’Oyapock		**Rodents** *Proechimys cuvieri*		5/72 (7%), DENV3 3/72 and DENV4 2/72
		[11,80]	2001–2007	16 (rodents) 39 (Marsupials) 19 (bats)	Camp du Tigre		**Rodents** *Mesomys hispidus* (1/5 DENV3) *Oecomys* spp. (13/28 DENV1 and DENV2) *Oryzomys megacephalus* (2/8 DENV1 and DENV2) *Proechymys cayennensis* (7/45 DENV and 2 and 3) **Marsupials** *Caluromys philander* (1/14 DENV1) *Didelphis marsupialis* (12/88 DENV2, 3 and 4) *Marmosops parvidens* (3/7 DENV3) *Marmosa murina* (16/96 DENV1, 2 and 3) *Micoureus demerarae* (4/45 DENV1, 2, 3 and 4) *Philander opossum* (3/37 DENV1 and 2) **Bats** *Artibeus planirostris* (14/42 DENV1, 2 and 3) *Carollia perspicillata* (5/63 DENV1 and 3)		16% (87/543) of the animals were infected on the site Distribution of DENV-1, -2, -3, and -4 was 41% (36/87), 20%, 33%, and 6%, respectively Chiroptera 4% (19/543), Rodentia 5%, and Marsupialia 7%
*Orthobunyavirus* *(Serogroup C)*	***Murutucu* virus**	[8]	1972–1975	NA	Cayenne Gallion	**Culicidae** *Anopheles peryassui Culex portesi*			
		[78]	1980	2	Montagne des Chevaux	**Culicidae** *Culex portesi*			
		[8,78]	1973	3	Cayenne			Three patients with fever, headache and myalgia.	
	***Oriboca* virus**	[8]	1975	1	Gallion	**Culicidae** *Culex portesi*			
	***Caraparu* virus**	[8]	1975	1	La Chaumière	**Culicidae** *Culex portesi*			
		[78]	1975	1	Gallion	**Culicidae** *Culex spissipes*			
		[78]	1974	1	Tonate	**Culicidae** *Mansonia titillans*			
		[8]	NA	NA	NA	*Limatus durhami*			
	**Group C undifferentiated (Muturucu, Oriboca, Caraparu)**	[78]	1973–1980	38	Gallion La Chaumière Matoury Montagne des chevaux Paramana Tonate Sinnamary Trou-Poissons	**Culicidae** *Aedes arborealis* *Anopheles peryassui* *Coquillettidia albicosta* *Coquillettidia venezuelensis* *Culex portesi* *Culex spissipes* *Mansonia titillans* *Trichoprosopon digitatum* *Trichoprosopon* sp. *Wyeomyia occulta*	**Marsupials** *Philander oppossum* **Rodents** Sentinel mice		
*Orthobunyavirus* *(Bunyamwera group)*	***Guaroa* virus**	[8,78]	1973	1	Gallion	**Culicidae** *Anopheles peryassui*			
	***Maguari* virus**	[8,78]	1978	3	Cité Jean-Marie Matoury Paramana	**Culicidae** *Wyeomyia* sp. *Culex portesi* *Wyeomyia aphobema* *Wyeomyia occulta*			
		[78]	1979	NA	Gallion	**Culicidae** *Anopheles* *nimbus*			
	***Wyeomyia* virus**	[8,78]	1975	1	Pont des Cascades	**Culicidae** *Wyeomyia occulta*			
		[78]	1977	1	Cabassou	**Culicidae** *Coquillettidia* *albicosta*			
		[8]	NA	NA	NA	**Culicidae** *Aedes taeniorhynchus* *Culex portesi* *Johnbelkinia longipes* **			
		[78]	1979	6	Paramana Gallion	**Culicidae** *Anopheles nimbus*			
*Orthobunyavirus* *(Guama group)*	**Guama group undifferentiated (Bimiti, Catu, Guama.)**	[8,78]	1972–1980	107 (mosquitoes)	Cité jean-Marie Gallion La Chaumière Matoury Montagne aux chevaux Paramana Rochambeau Sinnamary Tonate Trou-Poissons	**Culicidae** ***Coquillettidia venezuelensis*** ***Culex portesi*** *Anopheles braziliensis* *Anopheles darlingi* *Culex spissipes* *Culex taeniopus* ^$^ *Culex* sp. *Mansonia titillans* *Wyeomyia splendida* ^£^ *Wyeomyia occulta Psorophora ferox* *Sabethes undosus* *Trichoprosopon digitatum* ** *Trichoprosopon longipes* *Johnbelkinia longipes* **Phlebotominae** *Lutzomyia* sp.	**Rodents** Sentinel mice		
		[78]	1976–1978	19	Cogneau Inini La Chaumière Paramana Tonate Trou Poisson		**Birds** *Attila cinnamoneus* *Columba plumbea* *Elaenia chiriquensis* *Galbula dea* *Galbula galbula* *Myiarchus ferox* *Pitangus sulphuratus* *Platirynchus* sp. *Querula purpurata* *Ramphocelus carbo* *Tamnomanes ardesiacus* *Thruapis episcopus* *Thraupis palmarum* *Turdus fumigatus*		
		[78]	1976	5	La Chaumière Inini Paramana		**Marsupials** *Didelphis* *Marsupialis* **Rodents** *Proechimys guyanensis/ cuvieri*		
		[78]	1977	1	NA			Strain isolated from patient serum	
	***Guama* virus (Guama group)**	[8]	1974–1975	3	Gallion La Chaumière Paramana	**Culicidae** *Culex portesi* *Johnbelkinia longipes* **			
		[78]	1975–1976	2	Matoury		**Marsupials** *Didelphis marsupialìs*		
	***Bimiti* virus (Guama group)**	[8,78]	1972–1976	12	Gallion Iracoubo Matoury Paramana Trou Poisson	**Culicidae** *Culex* sp. *Culex portesi* *Culex taeniopus* ^$^ *Coquillettidia venezuelensis*			
		[8,78]	1972	1	Matoury		**Birds** *Galbula dea*		
	***Catu* virus (Guama group)**	[8,78]	1973–1976	4	Gallion Paramana Pont des Cascades	**Culicidae** *Culex portesi* *Culex spissipes*			
		[78]	1975	1	La Chaumière		**Marsupials** *Didelphis marsupialìs*		
	***Inini* virus (Simbu group)**	[45]	1973	1	Maripasoula		**Birds** *Pteroglossus aracari*		It shows some antigenic relationships with Mermet and Ingwavuma viruses
*Bunyamwera* like	***Itaporanga* virus (Phlebotomus group)**	[8,45]	1977–1978	2	Cité Jean-Marie¤Paramana	**Culicidae** *Culex albinensis* *Culex spissipes*			
		[78]	1977	1	Iracoubo		**Birds** *Nycticorax violacea*		
Ungrouped	***Aruac* virus**	[8]	NA	NA	NA	**Culicidae** *Coquillettidia albicosta* *Coquillettidia venezuelensis* *Culex* sp.			
	***Rochambeau* virus/*Paramana* virus**	[8,78]	1973	1	Paramana	**Culicidae** *Coquillettidia albicosta*			
			NA	1	Paramana	**Culicidae** *Culex portesi*			
		[78]	1974	1	Paramana		**Birds** *Tyrannus* *dominicensis*		

NA: data not available, Taxonomic names were ordered alphabetically by taxonomic groups and spelled exactly as published, and thus may not agree with currently accepted names. * *Anopheles mediopunctatus* has been confused with *Anopheles costai* and *Anopheles forattinii* in French Guiana [81]. Therefore, the viral isolations attributed to *An. mediopunctatus* could be attributed to *An. costai* or *An. forattinii*. ** *Johnbelkinia longipes* is the currently valid combination for this species [81]. ^$^
*Culex taeniopus* has been widely confused with *Culex. pedroi* on the coastal plain of French Guiana [82]. Therefore, the viral isolations attributed to *Culex. taeniopus* could be attributed to *Cx. pedroi*. ^£^
*Wyeomyia splendida* is currently the correct scientific name for this species [81].

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
