# Peer review of "Review on Main Arboviruses Circulating on French Guiana, An Ultra-Peripheric European Region in South America"

_viruses, 2023, doi:10.3390/v15061268_

Round 1
Reviewer 1 Report
This is a well written, comprehensive review of the situation regarding arboviruses in French Guiana. I found the review to be highly informative and a pleasure to read. I only have a few minor comments for the authors to consider.
1. The ICTV recommends that virus names should not be capitalized unless they contain a proper noun (a noun representing the name of a person or place). Thus the correct way to write these virus names is chikungunya virus, dengue virus, Zika virus, yellow fever virus, Mayaro virus, Oropouche virus, etc. Sometimes it may be necessary to do some digging to find the origin of a virus name and whether it contains any proper nouns or not.
2. Line 37, there is not a map shown in Fig. 1.
3. Line 105, define CRP.
Author Response
The requested corrections have been made on the attached file

Reviewer 2 Report
The authors carried out a broad and interesting review of the circulation of arboviruses in French Guiana, addressing various aspects, such as the biology of the main viruses found, the main outbreaks, the mosquito vectors, the difficulties and prospects for surveillance and control. In some points (highlighted below), the manuscript appears to be incomplete, as if it were a preliminary version, and must be corrected. Table 2 is especially informative and may help further work.
Line 27: Please, italicize “Aedes aegypti”.
Line 37: Figure 1, possibly a map, is missing.
Line 49: Please, insert references.
Lines 58-60: Please, insert references.
Line 72: Please use capital letter to “Alphavirus”.
Lines 83-98: Please insert references to support these phrases.
Lines 165: Please correct to “CHIKV”
206-207: The outbreak in Minas Gerais and Espirito Santo were also large. Maybe the authors can use “Brazilian Southeast states, such as Minas Gerais, Espírito Santo, Rio de Janeiro and São Paulo”
225: Please, provide references for YFV epidemics.
234: Please, correct to Covid-19
249: Please, capitalize Alphavirus
260-278: Please insert references to support these phrases.
293-301: Please insert references to support these phrases
312: Please, change to “Covid-19”
315-335: Please insert references to support these phrases
331: Please, use “Cx.” Instead of “Cu.” for Culex abbreviation.
336: remove the “track changes” mark
347: “MAYV”
350: What the authors means with “vector implantation”? Vector distribution? Vector introduction?
387: Please, remove the letter “a”.
413: Maybe the authors could also insert the distance in kilometers between the points.
421: Please, correct the reference “Juliano, 1998”.
437: Please, use “Ae.” instead of Aedes.
447: Please, remove “. :”
455: “(anymore?)” should be here?
470-473: This sentence appears to be unfinished due to the ellipses.
475: This sentence appears to be unfinished due to the ellipses.
Please, consider replacing “Indigenous” to “autochthonous” through the text.
522: Please, use “Ae.” instead of Aedes.
525: Please, use “Ae.” instead of Aedes.
Author Response

(The authors gave the same response as above.)

Reviewer 3 Report
The manuscript by Bonifay et al. describes anecdotal and current data on the prevalence and public health surveillance of seven arboviruses (Dengue, Yellow fever Zika, Chikungunya, Mayaro, Tonate, and Oropuche) in French Guyana. In addition, the authors depict the current challenges to controlling transmission and the potential epidemiological risks. I believe that research in South American arboviruses will benefit from an up-to-date review of the epidemiological situation in French Guyana; however, the authors need to condense and more precisely depict the information. In addition, significant grammatical (English) editing is necessary. I recommend this manuscript for publication after major revisions.
General recommendations:
· I suggest changing the title to reflect more precisely the content of the Review (i.e., Review on seven arboviruses circulating on French Guiana, an ultra-peripheric European region in South America)
· Italicize all species' scientific names.
· Use full genus when first introducing a mosquito species (Aedes aegypti instead of Ae. aegypti).
· Use the appropriate acronyms to refer to the Coronavirus disease 2019 (COVID-19) or the severe acute respiratory syndrome coronavirus 2 (SARS-CoV-2).
· Define all acronyms when first used (for example, CRP in line 105 and CMV in line 182)
Section-specific recommendations:
Abstract:
The abstract needs to be rephrased to easily summarize the content of the Review. See the example below.
French Guiana (FG), a French overseas territory in South America, is susceptible to tropical diseases, including arboviruses. The country's tropical climate supports the proliferation and establishment of invasive mosquito species that may act as vectors, making it challenging to control transmission. In recent years (please give a time range 2013-2023?), FG has experienced large outbreaks of imported arboviruses such as Chikungunya and Zika, as well as endemic arboviruses like Dengue, Yellow Fever, and Oropouche virus. Although Mayaro and Tonote viruses have a long history of sporadic cases, recent large outbreaks are not reported. The epidemiological surveillance of these seven arboviruses is challenging due to the diversity of the vector species. This article aims to summarize the current knowledge of these arboviruses in FG and discuss the challenges of arboviruses emergence and reemergence, and effective control.
Subsection 2.1 Dengue virus:
· Include the following information: Historically, which DENV serotypes circulate in FG? Is there any serotype more prevalent than the others? Is there any trend in the fluctuation between serotypes? Are the current epidemics exclusive of one serotype, or do several serotypes co-circulate simultaneously? Is there a serotype that is more prevalent in FG populations? Are there any reports about antibody enhancement due to past DENV infections?
· Please name which flavivirus or vaccines might represent cross-reactivity with anti-DENV IgM (lines 89-90).
· Clarify how the COVID-19 pandemic impacted DENV epidemiology. Specifically, which factors potentially contributed more to the increase in cases? What's it because people were more exposed to the mosquitoes at their hoses during a lockdown? Or because there was an increase in DENV surveillance to diagnose any febrile disease? Did any insecticide spread gaps occur during lockdowns?
· Regarding Dengue vaccines, please elaborate and use supporting information to explain why it is debatable that FG is a highly endemic area for DENV (line 113). Also, according to FDA and CDC, DENGVAXIA is not licensed for people over 16. There is insufficient data to show how well the vaccine works in that population. Please correct the statements in lines 113-114.
Subsection 2.2 Chikungunya virus:
· Include the following information: How is CHIKV diagnosis in FG (antibodies only or PCR)? What is the current CHIV situation? Are there any CHIKV cases (domestic or imported)? If there are no cases, please speculate why not?
Subsection 2.3 Zika virus:
· Include the following information: IS the PCR screening only in patience negative for DENV? Or is there any surveillance in susceptible comminates? Are there imported ZIKV cases? Has ZIKV become undetectable also in neighboring countries? If ZIKV is still circulating in neighboring countries, please hypothesize why it is not detected in FG.
Subsection 2.4 Yellow fever virus:
· Moving the first sentences of the second paragraph (lines 220-226) to start this section will add clarity and chronological order.
· Include which vaccine is in use in FG (YF-VAX (sanofi-pasteur), Stamaril (Aventis), others).
· Have other sylvatic Aedes species been implicated in the sylvatic transmission of YFV in FG?
· Are the mammalian species listed in the third paragraph (lines 241-245) suspected reservoirs of YFV? or dead-end hosts with previous infections?
Subsection 2.5 Mayaro virus:
· Other mosquito species from the genus Mansonia, Culex, Sabethes, Aedes, and Psorophora have also been implicated as vectors of MAYV. Please include this information and the corresponding citations accordingly.
Subsection 2.6 Tonate virus:
· Include the scientific name for the crested cacique bird (line 283).
· To maintain consistency, please add the common name and order for Trachops cirrhosis (as done in the previous sections).
· Is the TONV-like virus isolated in the U.S. a different serotype from the one circulating in FG? Or is it a closely related virus from another species?
Subsection 2.7 Oropuche virus:
· The correct abbreviation for the genus Culex is Cx. please replace “Cu. quinquefasciatus” with “Cx. quinquefasciatus” (line 331).
· Please use the correct scientific term ("Vectorial capacity" or "Vector competence") instead of "vectorial skills" (Lines 3330-332).
Subsection 2.8 Despite common characteristics, a great heterogeneity:
· This section is out of place and lacks appropriate transition and connection between ideas. Please reface the relevant content and incorporate it within 3.1-3.6 subsections.
Section 3. Challenges for the future:
· The overall content of the section needs to summarize better. It would also be beneficial to contrast past control and surveillance measurements with some recommendations on how to prevent disease transmission, new epidemics, and the emergence and re-emerges of new arboviruses.
· The authors need to keep in mind that most of the vector control measurements mentioned in these sections are only effective in urban settings where mosquito populations exploit artificial habitats for breeding. Those measurements don't extrapolate to sylvatic settings where mosquito habitats and behavior differ. Therefore, it is essential to discuss how antibody surveillance in human populations and specific examples of sentinel surveillance may help detect early epidemics of new or incoming arboviruses.
Conclusions:
· This section needs to be rephrased to easily highlight the main points of the Review. Individual sentences are confusing and lack appropriate transitions.
Tables and Figures:
· Table 1. please add Culicoides spp. as primary vectors of OROV. Also, replace "nb" with "number."
· Figure 4. Include the names of the main geographical points and cities mentioned within the text. This will help orientate international readers not familiar with FG territory.
Author Response

(The authors gave the same response as above.)

Round 2
Reviewer 3 Report
The manuscript by Bonifay et al. has significantly improved from earlier versions, and I recommend it for publication following minor revisions.
1. Please remain consistent on how to start each section. Use either the full virus name or the corresponding abbreviation. Since the virus name is part of the sub-title, I suggest starting the first paragraph with the acronym as done by most of the viruses (i.e., Line 72: DENV & Line 165: CHICKV).
2. Line 117: italicized the mosquito species name, Aedes.
3. Used the correct designated acronym for COVID-19 (coronavirus disease 2019) (Lines 115, 120, 172, 353, 363).
4. Line 211: Please use the correct spelling, either Zika virus or ZIKV.
5. Lines 413-414: Please include relevant reference(s) for this claim.
6. Line 443: italicized the bacteria species name: Wolbachia.
7. Lines 483-494: Use full virus names or acronyms throughout the section consistently. Since the virus names have already been introduced, I suggest changing “Mayaro virus” to MAYV (Lines 484 & 486).
8. Line 508-509: The demarcation of Orthobunyavirus species is difficult (thus lack of ICTV placement) due to reassorting variants and the lack of biochemical characterization of most isolates. Orthobunyavirus species are defined mainly by serological criteria (cross-neutralization and cross-hemagglutination-inhibition tests) or by a 10% difference in amino acid sequences of the N proteins. The author's ICTV note on Murutucu and Inini viruses is not relevant to the information presented in this section and does not add any epidemiological significance; therefore, I consider it should be removed.
9. Line 509: Use complete genus when first introducing a mosquito species, “Culex portesi”. Also, make appropriate corrections in Table 2 notes.
10. Line 523-524: not sure what “GF” means. Also, the idea in parenthesis must be further developed instead of “….”.
11. Table 2: The “virus” column should contain only the viruses’ names. Please use the “comment” column to define or better explain what “n=15 from 1974 to 1980” & “n=5 from 1973 to 1977” mean for Cabassou virus and Una virus, respectively.
12. Table 2 notes “*” and “$”: Unless the authors have compelling information (molecular data) on the corresponding virus isolation and the correct vector species taxonomy, please replace “should” with “could”. Sentences should read as follows: “*Anopheles mediopunctatus has been confused with Anopheles costai and Anopheles forattinii in French Guiana [83]. Therefore, the viral isolations attributed to An. mediopunctatus could be attributed to An. costai or An. forattinii” & “$ Culex taeniopus has been widely confused with Culex pedroi on the coastal plain of French Guiana [84]. Therefore, the viral isolations attributed to Cx. taeniopus could be attributed to Cx. pedroi.”
13. Table 2 notes “**” and “₤”: not sure what a “valid combination” is. Maybe, but not sure; the authors are refereeing to a “correct scientific name”.
Author Response
Hello,
Thank you for your involvement in the review of the article. The requested corrections have been made.
Sincerely
